# Prevalence and Predictors of Food Insecurity among Students of a Spanish University during the COVID-19 Pandemic: FINESCOP Project at the UPV/EHU

**DOI:** 10.3390/nu15081836

**Published:** 2023-04-11

**Authors:** Raquel González-Pérez, Laura García-Iruretagoyena, Naiara Martinez-Perez, Nerea Telleria-Aramburu, Saioa Telletxea, Sonia Padoan, Liv Elin Torheim, Marta Arroyo-Izaga

**Affiliations:** 1Department of Pharmacy and Food Sciences, University of the Basque Country UPV/EHU, 01006 Vitoria-Gasteiz, Spain; reiiich@hotmail.com (R.G.-P.); lgarcia.ir@gmail.com (L.G.-I.); nerea_telleria@hotmail.com (N.T.-A.); 2Department of Nursing I, University of the Basque Country UPV/EHU, 48940 Leioa, Spain; naiara.martinez@ehu.eus; 3BIOMICs Research Group, Microfluidics & BIOMICs Cluster UPV/EHU, Lascaray Research Centre, University of the Basque Country UPV/EHU, 01006 Vitoria-Gasteiz, Spain; 4Department of Social Psychology, Research Group in Social Psychology, University of the Basque Country UPV/EHU, 20018 San Sebastián, Spain; saioa.telletxea@ehu.eus (S.T.); sonia.padoan@ehu.eus (S.P.); 5Department of Nursing and Health Promotion, Faculty of Health Sciences, Oslo Metropolitan University (OsloMet), 0130 Oslo, Norway; livtor@oslomet.no; 6Bioaraba, BA04.03, 01006 Vitoria-Gasteiz, Spain

**Keywords:** food insecurity, university students, coronavirus, socioeconomic factors, risk factors

## Abstract

Research related to food insecurity (FI) among European university student populations is currently limited, especially the studies carried out during the COVID-19 pandemic. The present study aimed to assess the prevalence and identify possible predictors of FI among students from a Spanish public university, the University of the Basque Country UPV/EHU, during the COVID-19 pandemic. A cross-sectional observational study design was used, in which a total of 422 students completed an online survey. Results were weighted according to age and field of education. Binary logistic regressions adjusted by sex, age, and campus were applied to identify FI predictors. FI in 19.6, 2.6, and 0.7% of the population was mild, moderate, and severe, respectively. The three main predictors of FI were a decrease in the main source of income (OR, 2.80; 95% IC, 2.57–3.06), not receiving scholarships during the pandemic (OR, 2.32; 95% IC, 2.18–2.47), and living arrangements before the pandemic (not living with parents/relatives) (OR, 2.03, 95% IC, 1.89–2.18). This study found a high prevalence of FI among the students surveyed, and the strongest predictors of this FI were related to socioeconomic status. A robust and comprehensive policy response is recommended to mitigate FI in this population.

## 1. Introduction

Food insecurity (FI) is defined as limited or uncertain access to nutritionally adequate, safe, and acceptable foods that can be obtained in socially acceptable ways [1]. Experiences with FI can refer to insufficient access to food and being unable to afford to buy more, having anxiety about affording meals or eating a poor-quality diet as a result of limited financial ability [1]. The coronavirus disease (COVID-19) pandemic has had a profound negative impact on health and economic well-being in the world. In addition to the loss of life associated with COVID-19, the measures taken by governments to reduce the virus’s spread have resulted in large increases in unemployment, FI, and hunger [2].

According to the Food and Agriculture Organization of the United Nations (FAO), FI has increased globally since mid-2014, with nearly one in ten people living in this condition [3]. Moreover, before the pandemic, 11% of US households were classified as food-insecure [4]. In Europe, studies indicated that FI (measured by an inability to afford meat or a vegetarian equivalent) was 11% in 2012. FI levels decreased between 2005 and 2009, when they were at their lowest (9%), but have been increasing since then [5]. A recent study conducted during the COVID-19 pandemic in Spain indicated that the level of FI, using the FAO Food Insecurity Experience Scale (FIES) [6], has rapidly risen, especially in households in which several family members live together, and those in which some of the family members are in precarious working conditions [7].

College and university students were also considered to be a high-risk group for FI in various countries. Two recent literature reviews showed that about one in three students experienced FI before the COVID-19 crisis [8,9]. These reviews mainly included studies from the USA, Australia, and Africa. In general, food-insecure college and university students were more likely to be younger [10], Black or Hispanic [11,12], low-income or unemployed [12], housing-insecure, and receiving financial aid [13]. An early report from data collected during the COVID-19 crisis suggested that FI has risen above pre-pandemic levels among college and university students [14]. The main factors identified that have placed students at a high risk for FI during the COVID-19 pandemic in the USA have been the following: (1) higher unemployment rates affecting both students and their parents [15]; (2) the cost of goods (e.g., board) and services (tuition and room) during spring of 2020 that students had to pay for, but could not use due to the campus shutdowns because of COVID-19 [16]; (3) the need to relocate when universities and colleges shut down [17]; (4) the poor ability of students to plan, shop for, prepare and cook nutritionally balanced meals [18] (the closure of campus dining halls and cafeterias forced many college and university students to purchase and prepare meals on their own); (5) the limited access to community resources by college/university students (since it is supposed that they are economically supported by their parents) [19].

To date, FI among college and university students has received less attention in European countries than elsewhere. To our knowledge, research related to FI among European college/university student populations is currently limited, both before the pandemic [20] and in the context of the pandemic [21,22,23]. Nevertheless, there is direct, as well as indirect, evidence that FI and its attendant consequences are present in European universities. Given the similarities in food security between the USA and Europe, previous research can offer numerous insights into the causes and consequences of FI in Europe and possible directions to address them through measurement and public policy [24].

To address these gaps in the literature, the present study aimed to assess the prevalence of FI, and identify possible predictors of FI, such as demographic, socio-economic, and educational variables, as well as weight status, in students from a Spanish public university, the University of the Basque Country UPV/EHU, during the COVID-19 pandemic. Findings from this project will support the development of evidence-based campus initiatives and policies to address student malnutrition and financial challenges, in order to mitigate FI during and after the COVID-19 pandemic.

## 2. Materials and Methods

### 2.1. Study Design

This study is part of the Food Insecurity among European University Students during the COVID-19 Pandemic (FINESCOP) project, an observational, cross-sectional investigation of students enrolled in European universities, which aimed to better understand the vulnerable and challenging situation that students were experiencing during the COVID-19 pandemic. Initially, 61 institutions from sixteen countries were invited to participate in FINESCOP, but finally, eleven universities (from ten countries) participated in this consortium. The UPV/EHU, a public university located in the north of Spain, is one of the participating universities, as well as the study coordinator, together with Oslo Metropolitan University—OsloMet. This manuscript presents part of the results registered at the UPV/EHU, whose faculties and schools are distributed among three campuses: Álava/Araba, Bizkaia, and Gipuzkoa. 

### 2.2. Participant Recruitment and Enrollment

Subjects were eligible if (i) they were undergraduate or postgraduate students enrolled in the universities participating in FINESCOP during the study period, (ii) they were 18 years or older, and (iii) had access to the internet, since they had to complete an online questionnaire. No student was excluded, following the example of previous studies [14], in order to compare results. 

The sample size was estimated using the Epidat 3.0 program [25], considering the total number of students enrolled at each university during the 2020/21 academic year (information source: data published or provided by the Vice-Rectorate or Directorate of each university) and the prevalence of FI recorded in a previous similar study [14]. For this purpose, a precision level of 5%, a confidence interval of 95%, and a *p*-value of 0.05 were established. The estimated average sample size per university was 342 students. Appendix A displays the findings of other studies on the prevalence of FI among students of higher education, before and during the COVID-19 pandemic. At the UPV/EHU, the recruitment and data collection were conducted between December 2021 and January 2022. A timeline of the epidemiological situation and the state of Basque Country’s response to the COVID-19 pandemic and our survey administration is provided in Figure 1. The survey was advertised on all three campuses through the notice board (EHUTaula) and the website of the Sustainability Directorate of the UPV/EHU. Participants could complete the survey only if they consented to participate in this study on the first page. Participation was anonymous and an incentive of eight 50€ gift cards was used to encourage participation. 

A total of 422 students from the three campuses of the UPV/EHU completed the survey, with a participation rate of 1.0%. Qualtrics software (Provo, UT, USA) was used to fill out the questionnaire. The complete online questionnaire was designed to be completed in approximately 20 min on a computer or mobile device. Appendix A provides information on the population of the UPV/EHU, the theoretical sample achieved, the real sample obtained, the participation rate by field of education and age, and the weighting coefficient assigned to each participant. No students from the Services field participated in the survey, so it cannot be weighted for this field. The conduct and reporting of this work complies with STROBE-nut guidelines [26] (Appendix A).

### 2.3. Development of the Questionnaire

The questionnaire was developed in English through collaboration with all the partners in the FINESCOP consortium—who participated in the selection and consensus of the measurement variables and tools. The selection of variables to be studied was carried out regarding the experiences of other large multicenter studies, e.g., IDEFICS [27], and studies on FI previously conducted by partners of the FINESCOP consortium [28] and by other researchers [14,29]. 

For the evaluation of FI, the FIES [6] was used, which consisted of eight questions regarding people’s access to adequate food, and which could be easily integrated into several types of population surveys. At the UPV/EHU, we used the Spanish-translated FIES version for the Spanish population, available at the website: https://www.fao.org/in-action/voices-of-the-hungry/using-fies/en/ (accessed on 1 January 2021) (FIES survey module translations.zip).

In addition, the compulsory questionnaire included the following demographic variables: participant’s age, sex, birthplace, and migration status (if applicable); socio-economic variables: employment, income, living arrangements, participation in food assistance programs or other types of food assistance, and parents’ educational level; educational variables: campus, field of education, study level, academic year, teaching modality (campus attended, online only or blended), and scholarships; weight status (body mass index (BMI) calculated from self-reported weight and height data). Scholarships were considered an educational variable, because academic performance, in addition to financial income, was taken into account in their granting. 

To better understand the impact of some socio-economic factors, in particular, employment status, income, living arrangements, and scholarships, on FI, the questions referred to before the COVID-19 pandemic and the changes experienced since the beginning of the pandemic. Moreover, it was asked if the changes in employment status, income, and living arrangements were due to the pandemic situation. In those cases, in which respondents reported an increase or decrease in the number of working hours, they were asked to indicate the number, and in the case that respondents reported an increase or decrease in income, they were asked to detail the percentage of this change. The weight status of the subjects was classified according to their BMI using the World Health Organization criteria [30].

All the questions related to demographic, socio-economic, and educational variables were taken from the questionnaires developed and used by Owens et al. [14] and Mahy [31], except the questions related to the migration status [32] and parents’ educational level [33]. The tools that had not been previously validated in languages other than English were translated into the local languages using the parallel translation/double translation method [34]. In the case of the UPV/EHU, they were translated into Spanish and Basque. The participants also answered questions about the impact of the pandemic on aspects related to academic performance, health, and lifestyle. The FINESCOP questionnaire includes a total of 70 items, and eight of them involve open-ended answers. 

### 2.4. Questionnaire Piloting

A pilot study was carried out in five of the eleven participating universities in the FINESCOP project (including the UPV/EHU) before the approval of the final version of the questionnaire. The piloting involved three main phases: (1) early stages of questionnaire development, (2) structured field piloting, and (3) field implementation practice. The result of Phase 1 was a draft paper-based questionnaire in both English and the local language(s). The implementation of Phase 2 followed the Development Impact Evaluation’s survey piloting guide [35], and was structured in three parts: pre-pilot preparation, during piloting, and post-piloting feedback. Finally, the aims of Phase 3 were to practice the implementation of the survey, and to provide feedback and further practice to improve the implementation. The third phase of piloting included small convenient samples of college/university students. 

In addition, Phase 3 at the UPV/EHU allowed us to analyze the transcultural adaptation and validation of the questionnaire, through a sample of 37 students who responded to the Spanish version and 37 students who answered to the Basque version, before the actual distribution of the questionnaire. Internal consistency was evaluated for two of the subsections: socio-economic and educational factors. The Cronbach’s α results for socio-economic and educational items in the Spanish version were 0.83 and 0.79, respectively; and in the Basque version they were 0.78 and 0.77, respectively.

### 2.5. Quality Management of Data and Categorisation of the Open-Ended Answers

Unique subject identification numbers were used and attached to each register. Only completed questionnaires, i.e., those that answered all the questions in the compulsory sections, were included. Data preparation consisted of data cleaning based on traditional approaches, such as identification of inconsistencies, lie scores and response sets. For the data cleaning, the method proposed by Bonillo [36] was used. Regarding the categorization of the open-ended answers of the questionnaire, the manual inductive coding and flat coding frames method was applied [37]. At the UPV/EHU, the coding was done by two coders, where each coder checked the other’s coding to help eliminate cognitive biases. 

### 2.6. Statistical Analysis

Data were analyzed using IBM SPSS Statistics for Windows, version 28.0 (IBM Corp., Armonk, NY, USA). All descriptive statistics are presented as mean and standard deviation (SD) for continuous data, and as a percentage for categorical data. All results were weighted according to age and field of education, to ensure the representativeness of the UPV/EHU university students’ population, using weighting coefficients provided by the list of students enrolled in 2021/22 (data provided by Vice-rectorate of Digital Transformation and Communication of the UPV/EHU). 

Differences in continuous variables were assessed with the Mann–Whitney U test (the variables were not normally distributed, because the data were weighted). Categorical variables were analyzed using the Chi-squared test. The FIES variable was categorized into two groups: food-secure (total raw score of 0) and food-insecure (total raw score of 1–8) [6]. We combined the three categories of food insecurity (mild, moderate, and severe), due to the low prevalence of moderate and severe cases. 

To facilitate the analysis of associations between predictors and FI, some variables were dichotomized, as was in case of birthplace, migration status, employment, income, living arrangement, participation in food assistance programs/other types of food assistance, parental educational level, level of studies, and scholarships. Binary logistic regression adjusted by sex, age, and campus was used to identify the significant predictors, either as a cause or a consequence of FI. The results are presented in terms of odds ratios (OR), with a 95% confidence interval (CI). Multicollinearity between the independent variables was assessed by using variance inflation factors (VIF) before interpreting the final output, but no variables needed to be excluded.

The following independent variables referring to the situation of the participants were included in this analysis: socio-economic, and educational variables, and weight status. Participants with missing data for the covariates as a separate category were included. The reference categories were those reported in the literature to have a lower FI risk. All tests were 2-sided, and *p*-values less than 0.05 were considered statistically significant.

## 3. Results

### 3.1. Participant Characteristics

Table 1 shows the demographic and socio-economic characteristics of the sample. The average age of the participants was 22.7 (SD 5.5) years, most of them were women, born in Europe, and around 7% were immigrants. Immigrants had been residing in Spain for an average of 15.0 (SD 7.7) years.

The percentage of subjects who had experienced changes in employment status since the pandemic was 36.5%, and 22.9% of them stated that this change was due to the pandemic situation. Among those who responded that they had increased the number of working hours since the start of the pandemic; the average number of weekly hours increased was 14.1 (SD 4.0). Regarding changes in the main source of income, 27.6% had experienced such changes, both increases (13.3%) or decreases (14.3%), and 30.3% of these changes were related to the pandemic situation. The average increase in the main source of income was 60.9% (SD 37.2) and the average decrease in the main source of income was 41.7% (SD 28.1). Most of the participants lived with their parents or other relatives before the COVID-19 pandemic and were not working. The percentage of participants who had experienced changes in their living arrangements since the COVID-19 pandemic was 26.6%, and 14.3% of them stated that these changes were due to the pandemic situation. Finally, most of the parents or legal guardians of the participants had an (upper) secondary education or higher educational level.

Regarding educational characteristics, most of the students were undergraduates (Table 2). The three most frequent fields of education, ordered from highest to lowest, were: “Engineering, industry and construction,” “Business and law,” and “Health and wellness.” In addition, nearly four out of five students received face-to-face teaching. The prevalence of underweight and overweight/obesity was 6.4 and 12.4%, respectively. 

### 3.2. Prevalence and Predictors of FI during the COVID-19 Pandemic

A total of 22.9% of the population had experienced some form of FI during the COVID-19 pandemic, where 19.6% had experienced mild, 2.6% moderate, and 0.7% severe FI. Significant differences in age were found between the food-secure group (22.7 (5.8) years) and the food-insecure group (22.5 (4.1) years) (*p* < 0.001). Table 3 shows FI according to demographic, socio-economic, educational factors, and weight status. Participants who tended to have FI were males, immigrants, those who had worked before the pandemic, those whose employment status worsened during the pandemic, those who had suffered a decrease in the main source of income during the pandemic, those who did not live with their parents/other relatives, those whose parents/legal guardians had an educational level lower than tertiary education, and those who had lost their scholarship after the onset of the pandemic.

The regression analysis showed that the best predictors of FI were a decrease in the main source of income, not receiving scholarships during the pandemic, and not living with parents/relatives before the pandemic (Table 4). Other variables that were also related to FI, although to a lesser extent, were loss of scholarship after the onset of the pandemic, working before the pandemic, worsening of the employment status during the pandemic, parents/legal guardians had lower than tertiary education, and being an undergraduate student (as opposed to postgraduate). It should be noted that most of those living with parents/relatives were not working before the pandemic. Among those who lived with their parents, 76.3% did not work, and 23.7% did, while, among those with other type of living arrangement, 55.2% did not work, and 44.8% did (*p* < 0.001).

However, participation in food assistance programs/other strategies and a decrease in other sources of income were associated with a lower risk of FI. Regarding the latter result, it is noteworthy that there was no significant difference in the percentage decrease of the other sources of income between those who were classified as food-insecure and food-secure (44.3 (5.0) vs. 51.1 (39.1), *p* = 0.05). However, the percentage of decrease in the main source of income was greater in those who were classified as food-insecure than in those who were food-secure (49.7 (30.5) vs. 37.1 (25.6), *p* < 0.001).

## 4. Discussion

This study sought to assess the prevalence and predictors of FI among university students from a Spanish public university, during the COVID-19 pandemic. We found that about one in four university students experienced FI during the pandemic. The high rates of FI reported in the current study appear to be driven by economic predictors, such as a decrease in the main source of income during the pandemic, and by other factors related, at least in part, to socio-economic status, such as not receiving scholarships during the pandemic, and living arrangements before the pandemic (not living with parents/relatives).

The prevalence of FI experienced by this population (22.9%) is consistent with previous research (19%−62.8%) conducted among a combination of under- and postgraduate students [11,14,38,39,40]. The prevalence of FI in our sample was comparable to the estimated U.S. college prevalence of 19% in 2019 [11], and that estimated among Portuguese university students (17.3%) during the pandemic [22]. However, it was considerably lower than the prevalence found by other previously mentioned studies carried out during the pandemic [14]. The discrepancy with data from the U.S.A. during the pandemic, could be due, in part, to differences in sample characteristics (e.g., age, ethnic groups, field of education, living arrangement, among others) and methodological factors, making direct comparisons difficult. For example, some of the studies carried out in college/university students have typically not included the use of a screener (e.g., the two-item Food Sufficiency Screener), using only the six-item U.S. Department of Agriculture Food Security Module (FFSM) to assess FI [38,39,40]. At the same time, our rates of FI were higher than those reported in the Spanish households (~13%) during the COVID-19 pandemic using the FIES scale [7]. In this regard, prior to the COVID-19 pandemic, studies had consistently showed that college students had higher rates of FI than non-student U.S. households [10,11,29]. 

Given the limitations associated with comparisons to prior work and the nature of our study design, it is unclear if FI in university students increased during the COVID-19 pandemic. However, given the impact of the prolonged COVID-19 pandemic on household’s FI in Spain [7] and in other low- and middle-income countries [41], and higher rates of FI reported by students whose main source of income decreased during the pandemic in the current study, we believe that FI may have increased among university students during the COVID-19 pandemic. In this regard, other authors have observed an important increase in the percentage of the American college students who experienced FI during the pandemic, with an increase of 22.6% in the percentage of students who became more food-insecure after the onset of COVID-19 [42]. 

In the current study, one characteristic that was associated with FI during the COVID-19 pandemic was the decrease in the main source of income. Approximately one in ten students reported that their income decreased during the COVID-19 pandemic; these students had 2.80 greater odds of being food-insecure compared to students who did not experience a decline in their main income. These findings were consistent with previous studies showing that low income is an important determinant of FI [12,39], and, at the same time, they were consistent with the association observed in the current study between FI and other socio-economic variables (scholarships, employment status, and educational level of parents/legal guardians). Specifically, in the present study, working before the pandemic and having suffered a worsening of employment status during the pandemic were associated with a higher risk of FI. In this sense, Patton-López et al. [12] have also observed among students, that being employed was associated with FI with a similar increased risk (OR, 1.73; 95% CI, 1.04–2.88) as that described in the present study (OR, 1.58; 95% CI, 1.45, 1.73), although the percentage of students combining studies with work at the sample studied by Patton-López et al. [12] was higher than that obtained at the UPV/EHU (50.3% vs. 26.7%). Owens et al. [14] have also found that students whose current employment was directly affected by the COVID-19 pandemic were more likely to be food-insecure. 

In any case, the impact of the COVID-19 pandemic on unemployment or lack of resources and FI is not exclusive to college/university students [7,15]. It is likely that the pandemic also affected the parents or legal guardians of the surveyed students and, consequently, the FI of households. In fact, in a recent study on FI among Portuguese university students [22], it was found that households with a high number of unemployed members were more likely to be food-insecure. In the Spanish population, unemployment increased from 13.7% in February 2020 to 15.8% in June 2020 [43], with an increase in FI of 1.4% [7]. This rise in unemployment and FI was met with increased unemployment benefits, expanded food assistance programs, and other types of aid for disadvantaged people. However, many university students were not eligible for these emergency assistance programs that brought food and financial assistance to struggling people during the pandemic, because they were considered dependent on their parents’ tax returns. In addition to eligibility requirements, other reasons for the low participation of students in food assistance programs may have been application burden and stigma [44]. Indeed, despite the fact that about one in four university students experienced FI during the COVID-19 pandemic, less than 1% of students reported current participation in food assistance programs, while 2.7% turned to their parents/family/friends for food assistance or used other strategies for improving their access to food. Therefore, as other authors have previously pointed out [45], student-centered intervention strategies are necessary. 

It should also be noted that the Basque Government provides scholarships to students to pay for tuition and transportation if necessary. Having a low income is not the only requirement to receive this scholarship, and academic requirements must also be met. It may be that the decrease in academic performance justified the loss of scholarships during the pandemic, and that such a loss is related to the FI. However, no data are available in the present study to confirm whether the decline in academic performance was the reason for the loss of the scholarship. It should also be noted that the rising cost of college attendance has outpaced the financial aid that students receive [46]; coupled with the shifting demographics of university enrolment, these financial constraints have created a more economically vulnerable student body, and are key factors underlying the substantial socio-economic disparities in degree completion [47]. 

The socio-economic factors (including the social, economic, and cultural situation of the student’s family, or their own in case they are independent of their families) are some of the main reasons reported for dropping out prior to obtaining a higher education degree [47]. At the institution where the current study was conducted, the number of reintegrated assistance to students who could not afford tuition was more than double in the academic year in which the COVID-19 pandemic was declared (2019/20, *n* = 60) compared to the previous academic year (2018/19, *n* = 29) (data provided by the Vice-Rectorate for Student Affairs and Employability of the UPV/EHU), although this figure was reduced the following academic year (2020/21, *n* = 22).

Another characteristic associated with FI during COVID-19 among UPV/EHU students was the living arrangement before the pandemic. Compared to students who lived with parents or other relatives, students with other living arrangements had greater odds of being food-insecure. These findings agree with those reported by Owens et al. [14], who found that students who lived with parents or other relatives during the pandemic had lower odds of being food-insecure than students who lived alone. Our results were also in agreement with those of Bruening et al. [48], who found that students whose parents did not regularly send/purchase food for them were more likely to report FI. Although other studies have found an association between changes in housing and change in raw food security status score after the onset of COVID-19 [40], we did not find this association. In the current study, about one in four university students (26.6%) experienced some change in their housing situation, however, there was no difference in the food security situation between those who moved compared to those who did not move. This difference could be due to sociocultural aspects of the samples and/or differences in the sample sizes. In this sense, a high percentage of UPV/EHU students (69.1%) lived with their parents or relatives before the pandemic. These data agree with those of Spanish university students (56%) and indicate higher values than their European counterparts [49]. It is probable that this late emancipation of the UPV/EHU university students was conditioned by economic reasons and by the proximity of the campuses to the family residence. Living close to parents is a key factor for Spanish students when choosing a university [49].

This study has several strengths. To start with, we used a validated scale, the FIES, a direct measure of the experience of FI, which in turn can be analyzed together with indicators of its determinants and consequences to contribute to a more comprehensive understanding and inform more effective policies and interventions [3]. Moreover, demographic, socio-economic, and educational characteristics were collected, underlining associations with FI after the onset of COVID-19. Furthermore, the addition of questions related to changes in living arrangements, employment status, and incomes, as well as participation in any type of food assistance program, provided better insight into the main driving factors of FI during this pandemic. 

This study also has several limitations. First, this survey was cross-sectional and, as such, does not provide causality that the prevalence of FI observed in our sample was exclusively the result of the COVID-19 pandemic. However, given the predictive power of decreasing the main source of income and/or not receiving scholarships during the pandemic on FI in our survey sample, we believe that the COVID-19 pandemic is a likely contributor to the observed high rates of FI. 

In addition, a convenience sample from a university was used, which was not fully representative of all university students at the institution where the study was conducted. Survey respondents could choose whether to participate in the survey, which may have introduced selection bias. However, the authors weighted the sample according to age and field of education to account for these discrepancies. Thirdly, the survey response rates were low. A potential contributor to the low response rates in the current study could be the timing of our survey. The survey request was emailed to students at the end of the first quadrimester (mid-December) when many students were busy doing homework and/or studying for final exams. In any case, similarly, low response rates have been reported by other investigators assessing FI prevalence in college and university students [50].

Fourthly, the survey included questions asking students to recall their food security situation over the past 12 months and other related aspects before and after the onset of COVID-19. Students may have experienced recall bias when answering these questions. Moreover, all items were self-reported and may have been prone to recall and social desirability biases, as well as misinterpretation of the questions. Finally, there were other predictors of food security that were not assessed in this study, including nutrition and food literacy [10,18]. It is plausible that poor cooking skills, food shortages, and fear of grocery shopping may have contributed to FI among university students. As such, future studies are needed to explore how the COVID-19 pandemic impacted food accessibility, food selection, food preparation, and overall dietary quality in university students. Despite these limitations, this study yields important contributions to the field, particularly given the paucity of research to date on the issue of FI among European university students. 

## 5. Conclusions

In conclusion, this study found a high prevalence of FI among students surveyed at a southern European university during the COVID-19 pandemic. The strongest predictors of FI in this population were related to the socio-economic status, changes in the main source of income, receiving a scholarship during the pandemic, and student emancipation before the pandemic. 

A robust, comprehensive policy response is needed to mitigate FI in this population of university students. In this sense, it is important that policies are enacted at the university, community, and/or state level to alleviate the financial and health burden that university students are facing currently. Additional research is needed to ensure that interventions increase food security and ultimately improve their overall health. In addition, it is necessary to include the measurement of the levels of FI at the tertiary education level to monitor the state of access to sustainable food and how it is guaranteed, promoting solutions aimed at people.

## Figures and Tables

**Figure 1 nutrients-15-01836-f001:**
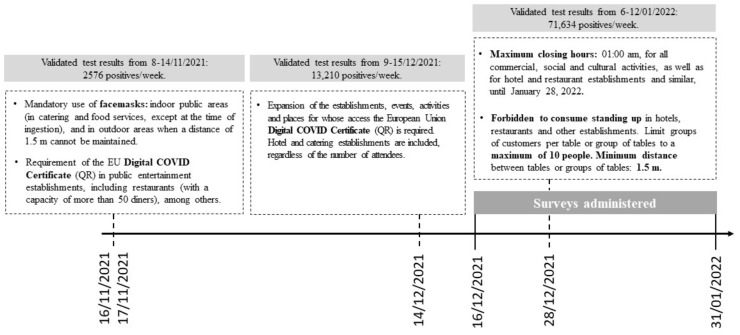
Survey collection timeline, epidemiological situation, and Basque Country mandates in response to the COVID-19 pandemic (information sources: Servicio Web del Departamento de Salud de Gobierno Vasco (Web service of the Health Department of the Basque Government). Transparencia sobre el Nuevo coronavirus (COVID-19): Datos COVID-19 en Euskadi (Transparency on the new coronavirus [COVID-19]: Data in the Basque Country). https://www.euskadi.eus/boletin-de-datos-sobre-la-evolucion-del-coronavirus/web01-a2korona/es/ (accessed on 1 February 2022); Servicio Web del Departamento de Salud de Gobierno Vasco (Web service of the Health Department of the Basque Government). Transparencia sobre el Nuevo coronavirus (COVID-19): Últimas medidas adoptadas en Euskadi (Transparency on the new coronavirus [COVID-19]: The latest measures adopted in Euskadi). https://www.euskadi.eus/normativa-de-medidas-excepcionales-adoptadas-por-el-nuevo-coronavirus-covid-19/web01-a2korona/es/ (accessed on 1 February 2022)).

**Table 1 nutrients-15-01836-t001:** Demographic and socio-economic characteristics of the sample: students of the University of the Basque Country UPV/EHU.

Variables	Survey Weighted ^1^ (*n* = 43,250)	
	*n*	% ^2^
Demographic variables		
Sex ^3^		
Female	28,860	68.5
Male	13,290	31.5
Place of birth		
Europe	41,944	97.0
Other	1306	3.0
Immigrant		
Yes ^4^	3178	7.3
Socio-economic variables		
Employment status before the pandemic ^5^		
Not employed and not looking for a job	26,277	62.1
Employed full-time	3933	9.3
Employed part-time	6254	14.8
Self-employed	1380	3.3
Not employed and looking for a job	2860	6.8
Other	1629	3.8
Changes in employment status during the pandemic ^6^		
No work before, nor do during the pandemic	22,183	51.5
No change at all	5092	11.8
No change in job, but working from home	5	0.1
Reduction of hours	1625	3.8
Increase of hours	1695	3.9
Loss of job	1775	4.1
Find the first job	5080	11.8
Not permitted to perform job due to public health restrictions	880	2.0
Change in job /new job	4679	10.9
Changes in the main source of income ^5^		
No change at all	23,713	72.4
Increase	4345	13.3
Decrease	4696	14.3
Changes in other sources of income ^5^		
No change at all	35,474	96.9
Increase	390	1.1
Decrease	761	2.1
Living arrangements before the pandemic ^6^		
Alone	1060	2.4
With roommates	7346	16.8
In college/university housing	2482	5.7
With parents or other relatives	30,188	69.1
With a partner	2156	4.9
With a partner and a child/children	460	1.1
Changes in living arrangements during the pandemic ^5^		
No change at all	31,478	73.3
Change to living alone	492	1.1
Change to living with roommates	3973	9.2
Change to living in college/university housing	2121	4.9
Change to living with parents or other relatives	3398	7.9
Change to living with a partner	1465	3.4
Changes to living with a partner and a child/children	44	0.1
Participation in food assistance programs or use other strategies ^5,6^		
No food aid	41,535	96.2
Food assistance program	511	1.2
Food assistance from parents/family/friends	671	1.6
Searching for edible food in waste containers	140	0.3
Other	339	0.8
Educational level of parents or legal guardians ^5^		
Pre-primary education or no education	929	2.2
Primary education	1763	4.1
Lower secondary education	2274	5.3
(Upper) secondary education	13,392	31.4
First stage of tertiary education	6889	16.2
Second stage of tertiary education	17,369	40.8

Note: ^1^ Weighted according to age and field of education; ^2^ Valid percentage; ^3^ The rest of participants answered “don’t know/don’t answer” or “non-binary”; ^4^ The remaining percentage answered “No”; ^5^ The rest of participants answered “don’t know/don’t answer”; ^6^ Multiple-choice questions.

**Table 2 nutrients-15-01836-t002:** Educational characteristics and weight status of the sample: students of the University of the Basque Country UPV/EHU.

Variables	Survey Weighted ^1^ (*n* = 43,250)	
	*n*	% ^2^
Educational variables		
Campus		
Álava/Araba	13,590	31.4
Bizkaia	15,876	36.7
Gipuzkoa	13,784	31.9
Field of education		
Education	4519	10.4
Arts and Humanities	5080	11.7
Social sciences, Journalism, and Information	5648	13.1
Business and Law	7618	17.6
Natural sciences, Mathematics and Statistics	3701	8.6
Information and Communication Technology	687	1.6
Engineering, Industry and Construction	10,077	23.3
Health and Wellness	5920	13.7
Level of studies and academic year		
Undergraduate	31,710	73.3
1st academic year	6880	21.7
2nd academic year	6966	22.0
3rd academic year	7232	22.8
4th academic year	9694	30.6
>=5th academic year	938	2.9
Postgraduate	11,540	26.7
Diploma	553	4.8
Master	3284	28.5
PhD	7703	66.7
Type of teaching received the last quadrimester		
Virtual or online	708	1.6
Face-to-face	32,267	74.6
Blended mixed	5555	12.8
No teaching received	4720	10.9
Scholarships ^3,4^		
Before the pandemic	14,855	29.4
During the pandemic	15,558	36.0
Weight status ^5^		
Underweight (BMI < 18.5 kg/m^2^)	2570	6.4
Normal weight (BMI = 18.5–25.0 kg/m^2^)	32,788	81.3
Overweight/obese (BMI >= 25.0 kg/m^2^)	4990	12.4

Abbreviation: BMI, body mass index. Note: ^1^ Weighted according to age and field of education; ^2^ Valid percentage; ^3^ The remaining percentage answered “No”; ^4^ Excluding those related to university exchange programs, for example, Erasmus; ^5^ The rest of participants answered “don’t know/don’t answer”.

**Table 3 nutrients-15-01836-t003:** Food (in)security status according to demographic, socio-economic, and educational characteristics, and weight status in the sample: students of the University of the Basque Country UPV/EHU.

Variables	Survey Weighted ^1^ (*n* = 43,250)				
	FS (*n* = 33,330)		FI (*n* = 9920)		*p* ^3^
	*n*	% ^2^	*n*	% ^2^	
Demographic variables					
Sex ^4^					**<0.001**
Female	23,024	79.8	5836	20.2	
Male	9519	71.6	3771	28.4	
Place of birth					**<0.001**
Europe	32,526	77.5	9418	22.5	
Other	804	61.6	502	38.4	
Immigrant					**0.003**
Yes	2381	74.9	797	25.1	
No	30,949	77.2	9123	22.8	
Sociodemographic variables					
Working before the pandemic ^5,6^					**<0.001**
Yes	9116	71.9	3571	28.1	
No	23,384	78.9	6261	21.1	
Changes in employment status during the pandemic ^5^					**<0.001**
Yes	9969	68.8	4529	31.2	
No	22,272	81.6	5031	18.4	
Worsening of employment status ^6,7^					**<0.001**
Yes	2236	64.6	1225	35.4	
No	30,005	78.3	8336	21.7	
Decrease in the main source of income ^5^					**<0.001**
Yes	3005	64.0	1691	36.0	
No	23,301	83.0	4757	17.0	
Decrease in other sources of income ^5^					**<0.001**
Yes	580	76.2	181	23.8	
No	27,870	77.7	7994	22.3	
Living arrangements before the pandemic ^5^					**<0.001**
With parents/relatives	23,727	78.6	6461	21.4	
Others ^8^	9603	73.7	3421	26.3	
Changes in living arrangements during the pandemic ^5^					0.117
Yes	8792	76.5	2701	23.5	
No	24,306	77.2	7172	22.8	
Participation in food assistance programs or					**<0.001**
use of other strategies ^5,9^					
Yes	736	64.5	405	35.5	
No	32,076	77.2	9459	22.8	
Educational level of parents or legal guardians ^10^					**<0.001**
<Tertiary education ^11^	13,200	71.9	5158	28.1	
Tertiary education ^12^	19,882	82.0	4376	18.0	
Educational variables					
Campus					**<0.001**
Álava/Araba	10,819	79.6	2771	20.4	
Bizkaia	12,462	78.5	3414	21.5	
Gipuzkoa	10,049	72.9	3735	27.1	
Level of studies of participants					**<0.001**
Undergraduate	24,516	85.4	7194	25.1	
Postgraduate	8814	76.4	2726	23.6	
Scholarships before the pandemic ^13^					**<0.001**
Yes	10,319	69.5	4536	30.5	
No	23,011	81.0	5384	19.0	
Scholarships during the pandemic ^13^					**<0.001**
Yes	10,491	67.4	5067	32.6	
No	22,839	82.5	4853	17.5	
Loss of scholarship after the onset of the					**<0.001**
COVID-19 pandemic ^13^					
Yes	2651	64.7	1449	35.3	
No	30,679	78.4	8471	21.6	
Weight status ^5^					
Underweight	2119	82.5	451	17.5	**<0.001**
Normal-weight	25,329	77.3	7459	22.7	
Overweight/obese	3926	78.7	1064	21.3	

Abbreviations: FI, food-insecure; FS, food-secure. Note: ^1^ Weighted according to age and field of education; ^2^ Valid percentage; ^3^ Significant *p*-values bolded; ^4^ The rest of participants answered “don’t know/don’t answer” or “no binary”; ^5^ The rest of participants answered “don’t know/don’t answer”; ^6^ Included: Employed full-time or part-time, self-employed, as well as seasonal and undeclared jobs (excluding unpaid work or internships); ^7^ Included: reduced work hours, loss of job or not permitted to perform job due to unsafe working conditions; ^8^ Included: with roommates, in college/university housing, with a partner and with a partner and a child/children; ^9^ Included: food assistance programs, food assistance from parents/family/friends, searching for edible food in waste containers and other strategies for improving access to food; ^10^ Highest educational level of parent or legal guardian who had achieved the highest educational level; ^11^ Included: post-secondary non-tertiary education, first stage of tertiary education, second stage of tertiary education; ^12^ Included: pre-primary education or no education, primary education, first stage of basic education, lower secondary education, second stage of basic education, (upper) secondary education and post-secondary non-tertiary education; ^13^ Excluding those related to university exchange programs, for example, Erasmus.

**Table 4 nutrients-15-01836-t004:** Predictors, either as a cause or as a consequence of food insecurity during the COVID-19 pandemic in the sample: students of the University of the Basque Country UPV/EHU.

Effect	OR (95% CI) ^1^	*p* ^2^
Decrease in the main source of income: yes vs. no (ref.)	2.80 (2.57, 3.06)	**<0.001**
Scholarships during the pandemic: no vs. yes (ref.)	2.32 (2.18, 2.47)	**<0.001**
Living with parents/other relatives before the pandemic: no vs. yes (ref.)	2.03 (1.89, 2.18)	**<0.001**
Loss of scholarship after the onset of the COVID-19 pandemic: yes vs. no (ref.)	1.67 (1.53, 1.82)	**<0.001**
Working before the pandemic: yes vs. no (ref.)	1.58 (1.45, 1.73)	**<0.001**
Worsening of employment status during the pandemic: yes vs. no (ref.)	1.36 (1.21, 1.52)	**<0.001**
Educational level of parents/legal guardians: lower than tertiary education vs. tertiary education (ref.)	1.23 (1.15, 1.31)	**<0.001**
Level of studies of participants: undergraduate vs. postgraduate student (ref.)	1.16 (1.06, 1.26)	**0.001**
Decrease in other sources of income: yes vs. no (ref.)	0.55 (0.45, 0.67)	**<0.001**
Participation in food assistance programs/other strategies: yes vs. no (ref.)	0.53 (0.45, 0.63)	**<0.001**

Abbreviations: OR, odds ratio; CI, confidence interval. Note: ^1^ Binary logistic regression with weighted data, adjusted by sex, age, and campus; ^2^ Significant *p*-values bolded.

## Data Availability

Data are to be made available only via a request to the corresponding author. Data will be provided only after the acceptance and signature of a formal data sharing agreement.

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
