# Peer review of "Prevalence and Predictors of Food Insecurity among Students of a Spanish University during the COVID-19 Pandemic: FINESCOP Project at the UPV/EHU"

_nutrients, 2023, doi:10.3390/nu15081836_

Round 1

Reviewer 1 Report

Thank you for the opportunity to review your manuscript. The paper presents an interesting thematic layer. Methodologically, the work stands at a high level. I do not make any changes to the manuscript. The authors could rethink the inclusion of sections on the strengths and limitations of the paper.

Author Response

Dear reviewer, 

We are greatly appreciative of your time and have made the requested revisions. Below you will find your comments and our responses.

Thank you for your time and consideration,

       Authors

-------------------------------------------------------------------------------

Reviewer #1: The authors could rethink the inclusion of sections on the strengths and limitations of the paper.

Authors’ answer: Information on the strengths and limitations of the paper was included in the Discussion section, between lines 436-444, and 445-471, respectively. 

Reviewer 2 Report

This is a well written and scientifically sound manuscript. There is a need for clarity of the scope of the population surveyed and potential limitations given timing of survey. Details follow.

It is not clear to me if the participants were from multiple universities or three campuses of one university.  If multiple universities were involved in the study, recommend changing the title: “a Spanish university” means one school. Instead, suggest: Prevalence and predictors of food insecurity among students from 61 Spanish universities during the COVID-19 pandemic: FINE-SCOP project

Methods: In the study design discussion, clarify population. I am confused—61 universities invited to participate in collaboration. Among those, 11 agreed for students to be surveyed, is that correct? However, this paper appears to report findings of for three campuses of one university, is that correct? If so, this needs to be more clearly articulated. If not, revise study design to reflect more accurately recruitment process and scope of population in this manuscript.

If multiple universities were involved in the study, recommend changing the title: “a Spanish university” means one school. Instead, suggest: Prevalence and predictors of food insecurity among students from 61 Spanish universities during the COVID-19 pandemic: FINE-SCOP project

Discussion: Add considerations about timing of survey. For more information, see table 4 in Reader J, Gordon B, Christensen N. Food Insecurity among a Cohort of Division I Student-Athletes. Nutrients. 2022 Nov 7;14(21):4703. doi: 10.3390/nu14214703. 

Line 21: Same edit as title, clarify university vs. universities.

Figure 1: Difficult to read. Recommend keeping test horizontal for those who do not print but read online only. Also, 12/28 date, suggest putting dotted line behind survey administered box. Otherwise, this is a very helpful figure for illustrating the current situation occurring in Spain during the survey time.

Author Response

Dear reviewer, 

We are greatly appreciative of your time and have made the requested revisions. Below you will find your comments and our responses.

Thank you for your time and consideration,

       Authors

------------------------------------------------------------------------------

Reviewer #2: It is not clear to me if the participants were from multiple universities or three campuses of one university.  If multiple universities were involved in the study, recommend changing the title: “a Spanish university” means one school. Instead, suggest: Prevalence and predictors of food insecurity among students from 61 Spanish universities during the COVID-19 pandemic: FINESCOP project

Authors’ answer: We appreciate the reviewer's comment. In order to clarify these issues, we introduced the following sentence in the manuscript:

  • Title: Prevalence and predictors of food insecurity among students of a Spanish university during the COVID-19 pandemic: FINESCOP project at the UPV/EHU.
  • 1. Study Design, lines 88-93: “This study is part of the Food Insecurity among European University Students during the COVID-19 Pandemic (FINESCOP) project, an observational cross-sectional investigation of students... Initially, 61 institutions from sixteen countries were invited to participate in FINESCOP, but finally, eleven universities (from ten countries) participated in this consortium.” Line 98-100: “This manuscript presents part of the results registered at the UPV/EHU, whose faculties and schools are distributed among three campuses: Álava/Araba, Bizkaia, and Gipuzkoa”.
  • 4. Questionnaire piloting, lines 188-189: “A pilot study was carried out in five of the eleven participating universities in the FINESCOP project (including the UPV/EHU) …”.

Reviewer #2: Methods: In the study design discussion, clarify population. I am confused—61 universities invited to participate in collaboration. Among those, 11 agreed for students to be surveyed, is that correct? However, this paper appears to report findings of for three campuses of one university, is that correct? If so, this needs to be more clearly articulated. If not, revise study design to reflect more accurately recruitment process and scope of population in this manuscript.

Authors’ answer: We included additional information in the Material and Methods section, lines 88-93: “This study is part of the Food Insecurity among European University Students during the COVID-19 Pandemic (FINESCOP) project, an observational cross-sectional investigation of university students... Initially, 61 institutions from sixteen countries were invited to participate in FINESCOP, but finally, eleven universities (from ten countries) participated in this consortium.”  Line 98-100: “This manuscript presents part of the results registered at the UPV/EHU, those faculties, and schools are distributed among three campuses…”.

Reviewer #2: If multiple universities were involved in the study, recommend changing the title: “a Spanish university” means one school. Instead, suggest: Prevalence and predictors of food insecurity among students from 61 Spanish universities during the COVID-19 pandemic: FINE-SCOP project

Authors’ answer: In order to clarify these issues, we included additional information in the title: “Prevalence and predictors of food insecurity among students of a Spanish university during the COVID-19 pandemic: FINESCOP project at the UPV/EHU”.

Reviewer #2: Discussion: Add considerations about timing of survey. For more information, see table 4 in Reader J, Gordon B, Christensen N. Food Insecurity among a Cohort of Division I Student-Athletes. Nutrients. 2022 Nov 7;14(21):4703. doi: 10.3390/nu14214703. 

Authors’ answer: We included a summary of the findings of other studies on the prevalence of food insecurity among students of higher education, before and during the COVID-19 pandemic, in the supplementary Table 1.

Reviewer #2: Line 21: Same edit as title, clarify university vs. universities.

Authors’ answer: We have reviewed it, and confirm that the correct word is “university”. Line 21: “…students from a Spanish public university, the University of the Basque Country (UPV/EHU) …”.

Reviewer #2: Figure 1: Difficult to read. Recommend keeping test horizontal for those who do not print but read online only. Also, 12/28 date, suggest putting dotted line behind survey administered box. Otherwise, this is a very helpful figure for illustrating the current situation occurring in Spain during the survey time.

Authors’ answer: We have made the changes suggested by reviewer #2 in Figure 1.

Reviewer #2: English language and style are fine/minor spell check required.

Authors’ answer: The manuscript has been checked by a native English-speaking colleague.
